# NOISE AGAINST NOISE: STOCHASTIC LABEL NOISE HELPS COMBAT INHERENT LABEL NOISE

**Pengfei Chen[1], Guangyong Chen[2]\*, Junjie Ye[3]\*, Jingwei Zhao[3], Pheng-Ann Heng[1,2]**
[1]The Chinese University of Hong Kong
[2]Shenzhen Institute of Advanced Technology, Chinese Academy of Sciences
[3]VIVO AI Lab
{pfchen, pheng}@cse.cuhk.edu.hk, {junjie.ye, jingwei.zhao}@vivo.com, gy.chen@siat.ac.cn

## ABSTRACT

The noise in stochastic gradient descent (SGD) provides a crucial implicit regularization effect, previously studied in optimization by analyzing the dynamics of parameter updates. In this paper, we are interested in learning with noisy labels, where we have a collection of samples with potential mislabeling. We show that a previously rarely discussed SGD noise, induced by stochastic label noise (SLN), mitigates the effects of inherent label noise. In contrast, the common SGD noise directly applied to model parameters does not. We formalize the differences and connections of SGD noise variants, showing that SLN induces SGD noise dependent on the sharpness of output landscape and the confidence of output probability, which may help escape from sharp minima and prevent overconfidence. SLN not only improves generalization in its simplest form but also boosts popular robust training methods, including sample selection and label correction. Specifically, we present an enhanced algorithm by applying SLN to label correction. Our code is released[1].

## 1 INTRODUCTION

The existence of label noise is a common issue in classification since real-world samples unavoidably contain some noisy labels, resulting from annotation platforms such as crowdsourcing systems (Yan et al., 2014). In the canonical setting of learning with noisy labels, we collect samples with potential mislabeling, but we do not know which samples are mislabeled since true labels are unobservable. It is troubling that overparameterized Deep Neural Networks (DNNs) can memorize noise in training, leading to poor generalization performance (Zhang et al., 2017; Chen et al., 2020b). Thus, we are urgent for robust training methods that can mitigate the effects of label noise.

The noise in stochastic gradient descent (SGD) (Wu et al., 2020) provides a crucial implicit regularization effect for training overparameterized models. SGD noise is previously studied in optimization by analyzing the dynamics of parameter updates, whereas its utility in learning with noisy labels has not been explored to the best of our knowledge. In this paper, we find that the common SGD noise directly applied to model parameters does not endow much robustness, whereas a variant induced by controllable label noise does. *Interestingly, inherent label noise is harmful to generalization, while we can mitigate its effects using additional controllable label noise.* To prevent confusion, we use stochastic label noise (SLN) to indicate the label noise we introduce. Inherent label noise is biased and unknown, fixed when the data is given. SLN is mean-zero and independently sampled for each instance in each training step. Our main contributions are as follows.

- We formalize the differences and connections of three SGD noise variants (Proposition 1-3) and show that SLN induces SGD noise that is dependent on the sharpness of output landscape and the confidence of output probability.

---

\*Corresponding to: gy.chen@siat.ac.cn, junjie.ye@vivo.com.
[1]https://github.com/chenpf1025/SLN

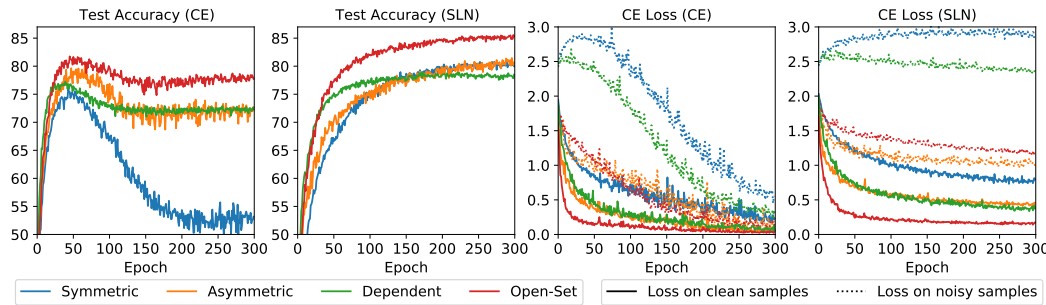

Figure 1: Test accuracy and training loss, averaged in 5 runs.

- Based on the noise covariance, we analyze and illustrate two effects of SLN (Claim 1 and Claim 2): escaping from sharp minima[2] and preventing overconfidence[3].

- We empirically show that SLN not only improves generalization in its simplest form but also boosts popular robust training methods, including sample selection and label correction. We present an enhanced algorithm by applying SLN to label correction.

In Fig. 1, we present a quick comparison between models trained with/without SLN on CIFAR-10 with symmetric/asymmetric/instance-dependent/open-set label noise. Throughout this paper, we use CE to indicate a model trained with standard cross-entropy (CE) loss without any robust learning techniques, while the standard CE loss is also used by default for methods like SLN. In Section 4, we will provide more experimental details and more results that comprehensively verify the robustness of SLN on different synthetic noise and real-world noise. Here, the test curves in Fig. 1 show that SLN avoids the drop of test accuracy, with converged test accuracy even higher than the peak accuracy of the model trained with CE. The right two subplots in Fig. 1 show the average loss on clean and noisy samples. When trained with CE, the model eventually memorizes noise, indicated by the drop of average loss on noisy samples. In contrast, SLN largely avoids fitting noisy labels.

## 2 RELATED WORK

### 2.1 SGD NOISE AND THE REGULARIZATION EFFECT

The noise in SGD (Wu et al., 2020; Wen et al., 2019; Keskar et al., 2016) has long been studied in optimization. It is believed to provide a crucial implicit regularization effect (HaoChen et al., 2020; Arora et al., 2019; Soudry et al., 2018) for training overparameterized models. The most common SGD noise is spherical Gaussian noise on model parameters (Ge et al., 2015; Neelakantan et al., 2015; Mou et al., 2018), while empirical studies (Wen et al., 2019; Shallue et al., 2019) demonstrate that parameter-dependent SGD noise is more effective. It is shown that the noise covariance containing curvature information performs better for escaping from sharp minima (Zhu et al., 2019; Daneshmand et al., 2018). On a quadratically-parameterized model (Vaskevicius et al., 2019; Woodworth et al., 2020), HaoChen et al. (2020) prove that in an over-parameterized regression setting, SGD with label perturbations recovers the sparse groundtruth, whereas SGD with Gaussian noise directly added on gradient descent overfits to dense solutions. In the deep learning scenario, HaoChen et al. (2020) present primary empirical results showing that SGD noise - induced by Gaussian noise on the gradient of the loss w.r.t. the model's output - avoids performance degeneration of large-batch training. Xie et al. (2016) discuss the implicit ensemble effect of random label perturbations and demonstrate better generalization performance. In this paper, we provide new insights by analyzing SGD noise variants and the effects, and showing the utility in learning with noisy labels.

---

[2]Around sharp minima, the output changes rapidly (Hochreiter & Schmidhuber, 1997; Keskar et al., 2017).
[3]The prediction probability on some class approaches 1.

## 2.2 ROBUST TRAINING METHODS

Mitigating the effects of label noise is a vital topic in classification, which has a long history (Ekholm & Palmgren, 1982; Natarajan et al., 2013) and attracts much recent interest with several directions explored. **1)** Malach & Shalev-Shwartz (2017); Han et al. (2018b); Yu et al. (2019); Chen et al. (2019a); Wei et al. (2020) propose *sample selection* methods that train on trusted samples, identified according to training loss, cross-validation or (dis)agreement between two models. **2)** Liu & Tao (2015); Jiang et al. (2018); Ren et al. (2018); Shu et al. (2019); Li et al. (2019a) develop *sample-weighting* schemes that aim to add higher weights on clean samples. **3)** Sukhbaatar et al. (2015); Patrini et al. (2017); Hendrycks et al. (2018); Han et al. (2018a) apply *loss-correction* based on an estimated noise transition matrix. **4)** Reed et al. (2015); Tanaka et al. (2018); Arazo et al. (2019); Zheng et al. (2020); Chen et al. (2020a) propose *label correction* based on the model's predictions. **5)** Ghosh et al. (2017); Zhang & Sabuncu (2018); Xu et al. (2019); Wang et al. (2019); Lyu & Tsang (2020); Ma et al. (2020) study *robust loss* functions that have a theoretical guarantee for noisy risk minimization, typically with the assumption that the noise is class-conditional (Scott et al., 2013; Natarajan et al., 2013). **6)** Chen et al. (2019b); Menon et al. (2020); Hu et al. (2020); Harutyun-yan et al. (2020); Lukasik et al. (2020) apply *regularization* techniques to improve generalization under label noise, including explicit regularizations such as manifold regularization (Belkin et al., 2006) and virtual adversarial training (Miyato et al., 2018), and implicit regularizations such as dropout (Srivastava et al., 2014), temporal ensembling (Laine & Aila, 2017), gradient clipping (Pascanu et al., 2012; Zhang et al., 2019; Menon et al., 2020) and label smoothing (Szegedy et al., 2016). **7)** One can combat label noise with *refined training strategies* (Li et al., 2019b; 2020; Nguyen et al., 2020) that potentially incorporate several techniques, including sample selection/weighting, label correction, meta-learning (Li et al., 2019b) and semi-supervised learning (Tarvainen & Valpola, 2017; Berthelot et al., 2019). Among these methods, regularization techniques are closely related to the essence of training networks, and studying robustness under label noise provides a new lens of understanding the regularization apart from the optimization lens.

## 3 METHOD

### 3.1 THE DIFFERENCES AND CONNECTIONS OF SGD NOISE VARIANTS

**Notations.** Let $\mathcal{D} = \{(x^{(i)}, y^{(i)})\}_{i=1}^n$ be a dataset with noisy labels. For each sample $(x, y)$, its label $y$ may be incorrect and the true label is unobservable. Let $f(x; \theta)$ be the neural network model with trainable parameter $\theta \in \mathbb{R}^p$. For a $c$-class classification problem, we have the output $f(x; \theta) \in \mathbb{R}^c$. We use a softmax function $S(f(x; \theta)) \in [0, 1]^c$ to obtain the probability of each class. The loss on a sample is denoted as $\ell(f, y)$. For classification, we use the cross-entropy (CE) loss by default. In parameter updates, a sample contributes $\nabla_\theta \ell(f, y)$ to the gradient descent. With SGD noise, the model is trained with a noisy gradient $\tilde{\nabla}_\theta \ell(f, y)$. Following the standard notation of the Jacobian matrix, we have $\nabla_\theta \ell \in \mathbb{R}^{1 \times p}$, $\nabla_f \ell \in \mathbb{R}^{1 \times c}$, $\nabla_\theta f \in \mathbb{R}^{c \times p}$, $\nabla_{\theta_i} f \in \mathbb{R}^c$ and $\nabla_\theta f_i \in \mathbb{R}^{1 \times p}$.

**Gaussian noise on the gradient of loss w.r.t. parameters.** The most common SGD noise is the spherical Gaussian noise directly added to the gradient to parameters (Neelakantan et al., 2015) as follow,

$$\tilde{\nabla}_\theta \ell(f, y) = \nabla_\theta \ell(f, y) + \sigma_\theta z_\theta, \tag{1}$$

where $\sigma_\theta > 0$, $z_\theta \in \mathbb{R}^{1 \times p}$ and $z_\theta \sim \mathcal{N}(0, I_{p \times p})$.

**Gaussian noise on the gradient of loss w.r.t. the model output.** Taking a step further, HaoChen et al. (2020) study SGD noise induced by label noise on a quadratically-parameterized regression model, whereas for classification, they add mean-zero noise to $\nabla_f \ell(f, y)$ as follow,

$$\tilde{\nabla}_\theta \ell(f, y) = (\nabla_f \ell(f, y) + \sigma_f z_f) \cdot \nabla_\theta f, \tag{2}$$

where $\sigma_f > 0$, $z_f \in \mathbb{R}^{1 \times c}$ and $z_f \sim \mathcal{N}(0, I_{c \times c})$.

**Noise induced by SLN.** The label perturbation is a common technique (Xie et al., 2016), while we provide new insights by analyzing the effects from the lens of SGD noise. Our SLN adds mean-zero Gaussian noise to the one-hot labels, where the noise is independently sampled for each instance in each training step.

$$\tilde{\nabla}_\theta \ell(f, y) = \nabla_\theta \ell(f, y + \sigma_y z_y), \tag{3}$$

where $\sigma_y > 0$, $z_y \in \mathbb{R}^c$ and $z_y \sim \mathcal{N}(0, I_{c \times c})$. Here, $\sigma_y z_y$ is the SLN on the label $y$.

In above Eq. (1-3), the standard deviation $\sigma$ is a hyperparameter. Since the SGD noise we introduce is i.i.d for each sample, we consider each single sample $(x, y)$ independently in the following propositions. For convenience, we use $f$, $S$ to indicate the model output $f(x; \theta)$ and the softmax output $S(f(x; \theta))$ on a sample. The proofs are presented in Appendix A.

**Proposition 1.** *Compared with Eq. (1), Eq. (2) induces noise $z \sim \mathcal{N}(0, \sigma_f^2 M)$ on $\nabla_\theta \ell(f, y)$, s.t., $M \in \mathbb{R}^{p \times p}$ and $M_{i,j} = (\nabla_{\theta_i} f)^T \nabla_{\theta_j} f$, $\forall i, j \in \{1, \cdots, p\}$. Note that the standard deviation of noise on the $i$-th parameter $\theta_i$ is $\sigma_f \|\nabla_{\theta_i} f\|_2$, where $\|\cdot\|_2$ denotes the $L_2$ norm.*

**Proposition 2.** *For the cross-entropy loss, compared with Eq. (2), Eq. (3) induces noise $z \sim \mathcal{N}(0, \sigma_y^2 M)$ on $\nabla_f \ell(f, y)$, s.t., $M \in \mathbb{R}^{c \times c}$, $M_{i,i} = c(S_i - 1/c)^2 + (c - 1)/c$ and $M_{i,j} = c \cdot S_i S_j - S_i - S_j$, if $i \neq j$. Note that the standard deviation of noise on the $i$-th entry is $\sigma_y \sqrt{c(S_i - 1/c)^2 + (c - 1)/c}$.*

**Proposition 3.** *For the cross-entropy loss, compared with Eq. (1), Eq. (3) induces noise $z \sim \mathcal{N}(0, \sigma_y^2 M)$ on $\nabla_\theta \ell(f, y)$, s.t., $M \in \mathbb{R}^{p \times p}$ and $M_{i,j} = (\frac{\nabla_{\theta_i} S}{S})^T \frac{\nabla_{\theta_j} S}{S}$, $\forall i, j \in \{1, \cdots, p\}$, where $\frac{\dot{}}{S}$ denotes the element-wise division. Note that the standard deviation of noise on the $i$-th parameter $\theta_i$ is $\sigma_y \left\| \frac{\nabla_{\theta_i} S}{S} \right\|_2$.*

## 3.2 THE EFFECTS OF SGD NOISE

Xie et al. (2016) discuss the effect of label perturbations as implicit ensemble. In this paper, based on Proposition 1-3, we show that *SLN induces SGD noise of high variance when the output landscape is sharp or the prediction confidence is high*. In this way, SLN *helps escape from sharp minima and prevents overconfidence*. It was discussed that flat minima generalize well (Hochreiter & Schmidhuber, 1997; Keskar et al., 2017; Neyshabur et al., 2017). Specifically, Achille & Soatto (2018) show that flat minima have lower mutual information between model parameters and training data, which leads to better generalization. The finding motivates several robust learning methods (Harutyunyan et al., 2020; Xie et al., 2020) and also supports our method. Moreover, preventing overconfidence can mitigate overfitting on noisy labels (Menon et al., 2020; Lukasik et al., 2020).

The SGD noise perturbs $\theta$ so that the training can not converge when the noise has high variance. Therefore, we derive Claim 1 and Claim 2 with justifications as follows.

- For the most common spherical Gaussian noise shown in Eq. (1), its standard deviation is a constant throughout training, independent of the landscape.

- For Eq. (2), Proposition 1 shows that the standard deviation of noise is $\sigma_f \|\nabla_{\theta_i} f\|_2$. $\|\nabla_{\theta_i} f\|_2$ can be very large around the sharp landscape, which means the SGD noise has high variance. The high variance makes the training difficult to converge, which helps escape from sharp minima. Note that for training without SGD noise, it can converge to sharp minima because $\theta$ always follows the direction of gradient descent, whereas the direction of noise is random.

- For our SLN in Eq. (3), Proposition 3 shows that the standard deviation of the SGD noise is $\sigma_y \|\nabla_{\theta_i} S / S\|_2$ with $\nabla_{\theta_i} S$ in the numerator. SLN similarly induces SGD noise with high variance around the sharp landscape. Hence we have Claim 1.

- Moreover, Proposition 2 shows that SLN induces SGD noise dependent on the confidence of $S$. The standard deviation of noise on an entry of $\nabla_f \ell(f, y)$ is minimized if $S_i = 1/c$, maximized if $S_i = 1$. Proposition 3 directly characterizes the equivalent noise on $\theta$, where $S$ is in the denominator (element-wise division). If $S$ is confident, s.t., the entropy $H(S) \to 0$, which means an entry of $S$ approaches 1 and others approach 0, then the variance will be high since there are small numbers in the denominator. Hence we have Claim 2.

**Claim 1.** *With SGD noise induced by Eq. (2) or Eq. (3), the training is difficult to converge when the output ($f$ or $S$) landscape is sharp.*

**Claim 2.** *With SGD noise induced by Eq. (3), the training is difficult to converge when the output ($S$) is overconfident, s.t. $H(S) \to 0$, where $H(\cdot)$ is the entropy.*

Fig. 2 shows visualizations of loss landscapes. The model trained with SLN converges to a flat minimum that has small SGD noise. More discussions on the convergence are presented in Appendix E.

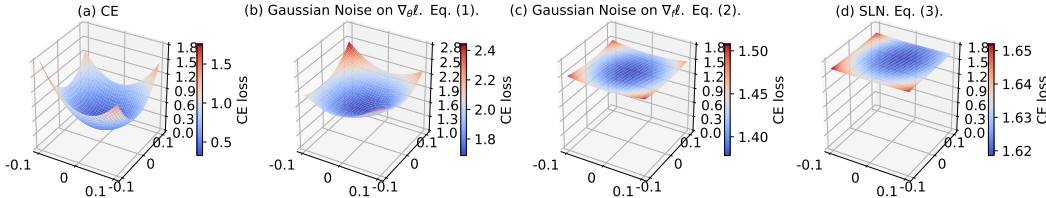

Figure 2: Loss landscapes around the local minima of converged models trained on CIFAR-10 with symmetric noise, visualized using the technique in Li et al. (2018). We show the z-axis on the same scale to compare the sharpness; and draw color bars separately to show the loss distribution around each minimum. (a): The model trained with CE converges to a **sharp minimum**. (b): Training with Eq. (1) yields a minimum with a higher loss, yet it is still sharp. (c)&(d): Consistent with our analysis, the model trained with Eq. (2) or Eq. (3) converges to a **flat minimum**.

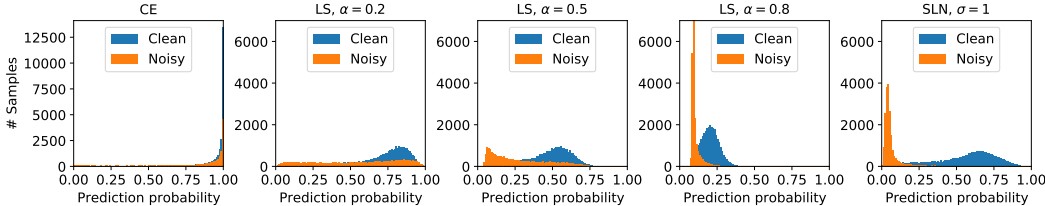

Figure 3: Samples density w.r.t. the prediction probabilities (the softmax outputs on the labeled class). SLN reduces overconfidence mostly on noisy samples, while clean samples are less affected.

In terms of escaping from sharp minima, there are previous works (Zhu et al., 2019; Daneshmand et al., 2018) that study the inheret noise in SGD, showing that the noise covariance containing curvature information performs better. They use the second-order approximation near the minima and apply integral to training steps to characterize the ability of escaping from sharp minima. In this paper, we study the noise induced by SLN and draw a more direct intuition between the noise covariance and the ability of escaping from sharp minima. Moreover, the dependency between the noise induced by SLN and the confidence of output probability further provides an intuition on avoiding overfitting under noisy labels. In terms of preventing overconfidence, we shall discuss label smoothing (LS) (Lukasik et al., 2020). It smooths the given one-hot label $y$ into a soft one $\tilde{y}$, s.t., $\tilde{y} = (1-\alpha)y + \alpha e/c$, where $e$ is an all-one vector and $\alpha > 0$. In this way, LS introduces a fixed and biased perturbation on the label, whereas SLN introduces dynamic and mean-zero perturbations. LS does not introduce noise in each training step, while SLN adaptively perturbs the parameter once the landscape is sharp or the prediction is overconfident. Hence, the robustness of SLN may not result from preventing overconfidence alone, but also escaping from sharp minima. In Fig. 3, we plot the sample density w.r.t. predictions on the labeled class, using CIFAR-10 with $40\%$ symmetric noise as an example. It shows that SLN does reduce overconfidence, while it mostly affects noisy samples.

## 3.3 A DISSECTION ON TRAINING SAMPLES

In this section, we show that SLN boosts popular robust training methods, including sample selection and label correction. Many methods, demonstrated to work well, select or add higher weight on small-loss samples (Han et al., 2018b; Jiang et al., 2018; Li et al., 2020), or use the model's predictions to correct noisy labels (Tanaka et al., 2018; Arazo et al., 2019). A warm-up phase is usually required to initialize the model before sample selection or label correction, yet the model will memorize noise if the warm-up phase is too long. With SLN, we can simply train the model until convergence. In Fig. 4, we compare converged models on CIFAR-10, where the model is trained with/without SLN. The detailed noise setting can be found in Section 4. We first sort training samples in ascending order of loss, then uniformly divide them into 1000 samples per interval, and finally obtain the number of four types of samples in each interval based on the correctness of the given label and the prediction. When trained with CE, there are many small-loss samples with incorrect labels (the blue region), then selecting small-loss samples is not reliable. The model

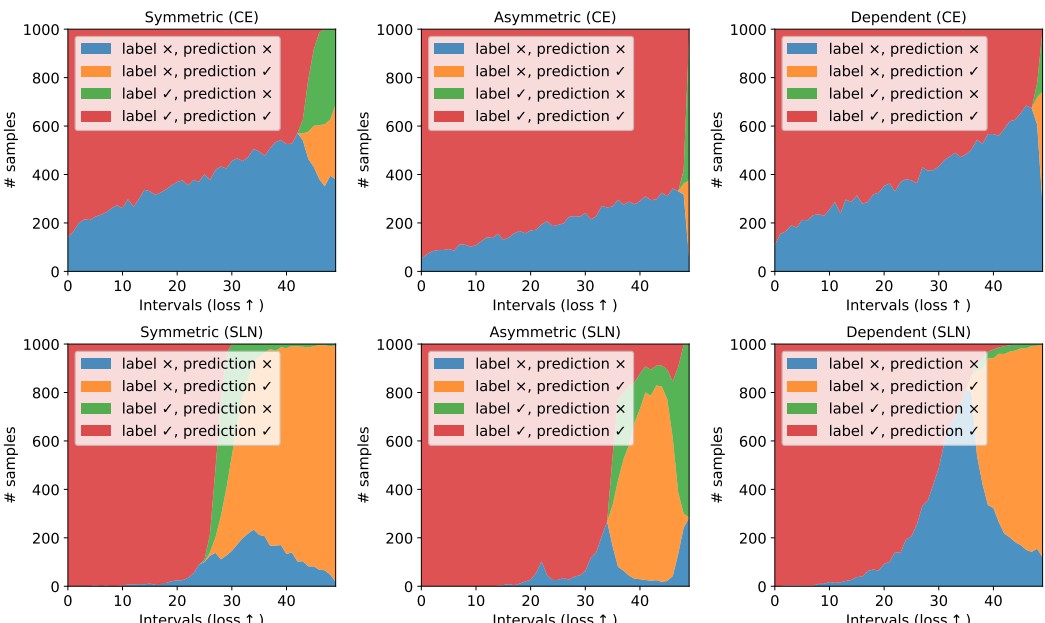

Figure 4: Training samples are sorted in ascending order of loss, uniformly divided into 1000 samples per interval, and dissected according to the correctness of the given label and the prediction.

trained with SLN largely addresses the issue. Moreover, SLN is suitable for label correction since it yields correct predictions for many originally incorrect samples (the orange region).

As a concrete example, we present an enhanced algorithm by applying SLN to label correction. With SLN, we train the model for sufficient epochs until convergence, without the need of carefully tuning a warm-up phase. Then we start label correction using $y_{correction} = \omega \cdot y + (1 - \omega) \cdot S$, where $S$ is the softmax prediction, $\omega \in [0, 1]$ is the weight obtained by normalizing the training loss, s.t., for the $i$-th training sample, $\omega_i = (\ell_i - \ell_{min})/(\ell_{max} - \ell_{min})$. More discussions on the label correction are presented in Appendix D.

## 4 EXPERIMENTS

### 4.1 EXPERIMENT SETUP

We comprehensively verify the utility of SLN on different types of label noise, including symmetric noise, asymmetric noise (Zhang & Sabuncu, 2018), instance-dependent noise (Chen et al., 2020a) and open-set noise (Wang et al., 2018) synthesized on CIFAR-10 and CIFAR-100 and real-world noise on Clothing1M (Xiao et al., 2015).

- Symmetric noise assumes each label has the same probability of flipping to any other class. We uniformly flip the label to other classes with an overall probability $40\%$.

- Asymmetric noise contains noisy labels flipped between similar classes. Following Zhang & Sabuncu (2018), on CIFAR-10, we flip labels between TRUCK→AUTOMOBILE, BIRD→AIRPLANE, DEER→HORSE, and CAT↔DOG, with a probability $40\%$; on CIFAR-100, we flip each class into the next class circularly with a probability $40\%$.

- Instance-dependent noise is challenging since the mislabeling probability should be dependent on each instance's input features (Xia et al., 2020; Chen et al., 2020a). We use the instance-dependent noise from Chen et al. (2020a) with a noise ratio $40\%$, where the noise is synthetized based on the DNN prediction error.

- Open-set noise contains samples that do not belong to any class considered in the classification task. Following Wang et al. (2018), we yield CIFAR-10 with open-set noise by randomly replacing $40\%$ of its training images with images from CIFAR-100.

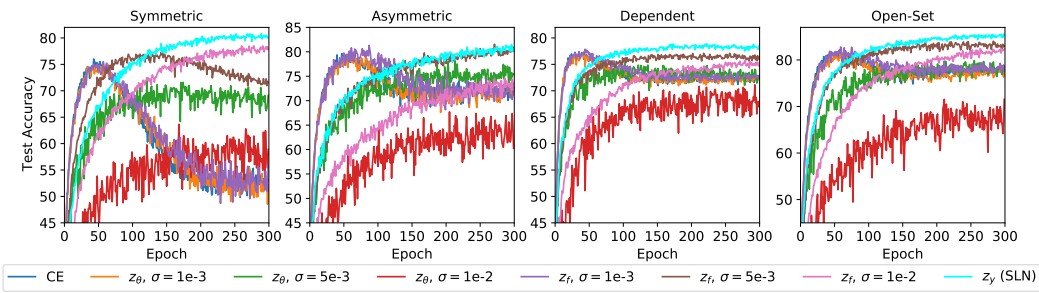

Figure 5: Performance of SGD noise variants on CIFAR-10. The accuracy is averaged in 5 runs.

- For real-world noise, we use the large-scale benchmark Clothing1M, which contains $1M$ training images with noisy labels from online shops.

We use the backbone wide ResNet-28-2 (Zagoruyko & Komodakis, 2016) on CIFAR-10 and CIFAR-100 and ResNet-50 on Clothing1M. More training details can be found in Appendix B. Apart from SLN, we introduce a momentum (MO) model (Tarvainen & Valpola, 2017) and label correction (LC), with which we get SLN-MO and SLN-MO-LC. Parameters of the momentum model ($\theta'$) is updated as a moving average of the the training model ($\theta$). At $t$-th training step, $\theta'^{(t)} = \alpha\theta'^{(t-1)} + (1 - \alpha)\theta^{(t)}$. We use $\alpha = 0.999$ in all experiments. For SLN-MO-LC, we have discussed in Section 3.3 that SLN is reliable when applied to label correction.

## 4.2 COMPARING SGD NOISE VARIANTS

We first show that compared with other variants, SLN stands out in mitigating the effects of label noise. The SGD noise variants have been formalized in Eq. (1-3), including $z_\theta$ directly added to $\nabla_\theta\ell$, $z_f$ on $\nabla_f\ell$ and our SLN ($z_y$). For SLN, the standard deviation is $\sigma = 1$ under symmetric label noise and $\sigma = 0.5$ in all other cases. For other SGD noise, the standard deviation is equally tuned and the performance under three different $\sigma$ is separately shown in Fig. 5. SLN significant improves the generalization, while other SGD noise variants can not achieve such impressive performance even if $\sigma$ is heavily tuned. It is worth noting that for all these variants, when $\sigma$ is too small, the model overfits noise since the training is similar to merely using CE loss; when $\sigma$ is too large, the model fails to fit the training data since the SGD noise is too high.

## 4.3 CIFAR-10 AND CIFAR-100

On CIFAR-10 and CIFAR-100, we compare with the following baselines: 1) standard cross-entropy (CE) loss, 2) Generalized Cross-Entropy (GCE) (Zhang & Sabuncu, 2018) loss, 3) Co-Teaching (Han et al., 2018b) that uses co-training and sample selection, 4) PHuber-CE (Menon et al., 2020) that uses gradient clipping and 5) label-smoothing (LS) Lukasik et al. (2020) that clips the label to be less confident before training. We use $5k$ noisy samples as the validation to tune hyperparameters, then train the model on the full training set and report the test accuracy at the last epoch. SLN simply requires tuning the standard deviation $\sigma$, which is tuned in $\{0.1, 0.2, 0.5, 1\}$. On CIFAR-10, the best $\sigma$ is 1 under symmetric noise and 0.5 otherwise; On CIFAR-100, it is 0.1 under instance-dependent noise and 0.2 otherwise. The label correction in SLN-MO-LC is applied in the last 50 epochs. The softmax prediction is converted into one-hot label in correction. We repeat each experiment 5 times.

The average test accuracy at the last epoch is reported in Table 1 and Table 2. To illustrate the influence of $\sigma$, an ablation study is presented in Fig. 6. In the tables, we mark the top-3 results in bold and present the average training time of each method, evaluated on a single V100 GPU. Without the momentum model and label correction, vanilla SLN achieves impressive test performance, which is consistent with results in Fig. 1. In Section 3.3, we have analyzed that SLN can boost popular robust training methods, based on a detailed dissection on DNNs' predictions. As expected, we obtain further improvement with SLN-MO and SLN-MO-LC. Notably, SLN, SLN-MO and SLN-MO-LC sweep the top-3 results in almost all cases and require low computational cost. We do not expect vanilla SLN to achieve state-of-the-art performance compared with many integrated methods. Still,

Table 1: Test accuracy (mean±std in 5 runs) on CIFAR-10. The Open-Set noise is generated by randomly replacing 40% images of CIFAR-10 with images from CIFAR-100. In Appendix C.1, Table 4 shows that SLN can improve many robust learning methods.

| Method | Symmetric | Asymmetric | Dependent | Open-Set | *Training time* |
|---|---|---|---|---|---|
| CE | 53.16±1.64 | 72.72±1.35 | 72.33±0.50 | 77.74±0.84 | *1.18±0.03h* |
| GCE | 79.27±0.54 | 70.90±2.98 | 73.05±0.24 | 81.56±0.41 | *1.19±0.03h* |
| Co-Teaching | **82.37±0.32** | 79.61±1.06 | 76.69±0.66 | 85.21±0.35 | *2.22±0.06h* |
| PHuber-CE | 80.58±0.34 | 73.01±1.28 | 72.70±0.24 | 81.62±0.52 | *1.20±0.04h* |
| LS | 66.59±1.25 | 66.96±2.02 | 72.79±0.27 | 77.06±0.37 | *1.20±0.02h* |
| SLN | 80.00±0.61 | **80.63±0.61** | **78.48±0.28** | **85.33±0.52** | *1.24±0.05h* |
| SLN-MO | **84.71±0.43** | **84.80±0.27** | **80.56±0.16** | **89.16±0.31** | *1.30±0.03h* |
| SLN-MO-LC | **87.00±0.27** | **87.85±0.41** | **81.76±0.16** | **88.47±0.22** | *1.31±0.04h* |

Table 2: Test accuracy (mean±std in 5 runs) on CIFAR-100.

| Method | Symmetric | Asymmetric | Dependent | *Training time* |
|---|---|---|---|---|
| CE | 31.63±0.83 | 36.26±0.65 | 55.59±0.43 | *1.22±0.04h* |
| GCE | 44.07±0.56 | 36.75±1.07 | 54.51±0.26 | *1.23±0.05h* |
| Co-Teaching | **53.05±1.11** | 39.78±0.52 | 47.06±0.44 | *2.23±0.07h* |
| PHuber-CE | 48.76±1.28 | 35.13±0.38 | 55.11±0.29 | *1.29±0.04h* |
| LS | 40.57±0.68 | 43.51±0.48 | 53.89±0.39 | *1.23±0.03h* |
| SLN | 50.24±0.41 | **44.43±0.19** | **57.47±0.30** | *1.28±0.04h* |
| SLN-MO | **56.57±0.38** | **50.59±0.35** | **61.24±0.27** | *1.32±0.02h* |
| SLN-MO-LC | **58.64±0.37** | **63.90±0.46** | **61.14±0.39** | *1.35±0.03h* |

SLN can be a promising option in the family of robust learning methods. In Appendix C.1, we show that SLN can improve many existing methods.

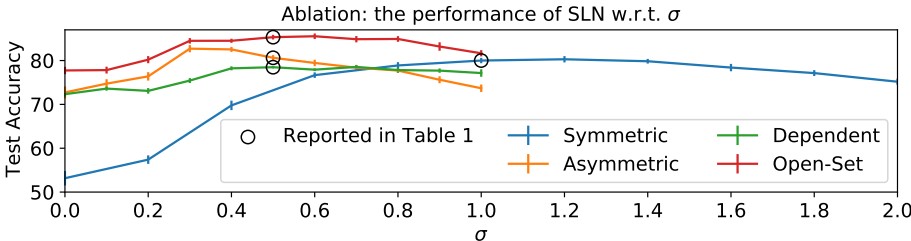

Figure 6: Test accuracy (mean±std in 5 runs) of SLN on CIFAR-10 w.r.t. $\sigma$. A small $\sigma$ results overfitting while a large $\sigma$ yields underfitting. In Appendix C.2, we visualize the embedding of overfitting/underfitting. For results reported in Table 1 and Table 2, following Zhang & Sabuncu (2018); Chen et al. (2020b), we use $5k$ noisy samples as the validation to tune $\sigma \in \{0.1, 0.2, 0.5, 1\}$.

## 4.4 CLOTHING1M

Clothing1M (Xiao et al., 2015) is a large-scale benchmark of clothing images from online shops with 14 classes, containing real-world label noise. It has 1 million noisy samples for training, $14k$ and $10k$ clean samples for validation and test. The number of images labeled as each class is unbalanced, ranging from 18976 to 88588 in the noisy training set. In previous works, some experiments are conducted by sampling a class-balanced training subset in each epoch (Li et al., 2020), while others directly train on the full training set (Patrini et al., 2017). Since the balanced training sampling itself affects the test performance, it is difficult to compare results across papers. Therefore, we

Table 3: Test accuracy (mean±std in 3 runs) on Clothing1M. The star* marks results copied from Patrini et al. (2017). The result of DivideMix (Li et al., 2020) is reproduced from its official implementation, which uses class-balanced training sampling.

| Training sampling | Standard | | Noisy-class-balanced | |
|---|---|---|---|---|
| Method | Test accuracy | *Training time* | Test accuracy | *Training time* |
| CE | 68.94* | - | 71.12±0.32 | *2.61±0.08h* |
| Forward | 69.84* | - | 71.28±0.27 | *2.74±0.05h* |
| Backward | 69.13* | - | 71.03±0.33 | *2.72±0.09h* |
| Co-Teaching | 70.19±0.28 | *15.06±0.33h* | 72.14±0.28 | *4.51±0.17h* |
| DivideMix | - | - | 73.81±0.41 | *18.78±0.32h* |
| SLN | 70.42±0.34 | *9.25±0.12h* | 72.95±0.31 | *2.73±0.05h* |
| SLN-MO | 71.15±0.21 | *9.31±0.09h* | 72.98±0.15 | *2.78±0.07h* |
| SLN-MO-LC | **72.61±0.23** | *11.59±0.23h* | **74.08±0.18** | *3.31±0.03h* |

make it clear here and conduct experiments in both setting, including the standard sampling and the noisy-class-balanced sampling. For the latter, in each epoch, 18976 instances per class are randomly sampled from the noisy training set. Other training details strictly follow the standard benchmark setting (Patrini et al., 2017), presented in Appendix B. We set the standard deviation of SLN as $\sigma = 0.2$. For SLN-MO-LC, the label correction is applied since the first epoch. Results are listed in Table 3, with the best result in bold and previous published results marked by a star. DivideMix trains two models in each run and we average their test accuracy, rather than using additional ensemble of two models. Our SLN outperforms many baselines. The variants SLN-MO and SLN-MO-LC further achieve higher test accuracy. Specifically, SLN-MO-LC achieves the best test accuracy in both settings. Our methods also stand out for training efficiency.

## 5 CONCLUSION

In this paper, we establish that SLN induces SGD noise dependent on the sharpness of output landscape and the confidence of output probability and analyze the effects of escaping from sharp local minima and preventing overconfidence. This partially explains the robustness of SLN under noisy labels since various works show that flat minima typically generalize well and preventing overconfidence helps mitigate overfitting on noisy labels. We empirically verify the robustness of SLN under various synthetic label noise and real-world noise. Moreover, we show that SLN boosts popular robust training methods, including sample selection and label correction. In particular, we justify that SLN can enhance existing methods based on a detailed dissection on training samples, then present a practical algorithm by applying SLN to label correction.

## ACKNOWLEDGMENTS

This work was supported by a grant from Research Grants Council of the Hong Kong Special Administrative Region (Project No. CUHK 14201620) and the National Natural Science Foundation of China (Project No. 62006219).

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

## A  PROOFS

**Proposition 1.** *Compared with Eq. (1), Eq. (2) induces noise $z \sim \mathcal{N}(0, \sigma_f^2 M)$ on $\nabla_\theta \ell(f, y)$, s.t., $M \in \mathbb{R}^{p \times p}$ and $M_{i,j} = (\nabla_{\theta_i} f)^T \nabla_{\theta_j} f, \forall i, j \in \{1, \cdots, p\}$. Note that the standard deviation of noise on the $i$-th parameter $\theta_i$ is $\sigma_f \|\nabla_{\theta_i} f\|_2$, where $\|\cdot\|_2$ denotes the $L_2$ norm.*

*Proof.* Dimension: $z_f \in \mathbb{R}^{1 \times c}, \nabla_\theta \ell \in \mathbb{R}^{1 \times p}, \nabla_f \ell \in \mathbb{R}^{1 \times c}, \nabla_\theta f \in \mathbb{R}^{c \times p}, \nabla_{\theta_i} f \in \mathbb{R}^c$.

For Eq. (2), the noisy gradient is

$$\tilde{\nabla}_\theta \ell(f, y) = (\nabla_f \ell(f, y) + \sigma_f z_f) \cdot \nabla_\theta f = \nabla_\theta \ell(f, y) + \sigma_f z_f \cdot \nabla_\theta f \quad (4)$$

The noise on $\nabla_\theta \ell(f, y)$ is $z = \sigma_f z_f \cdot \nabla_\theta f \in \mathbb{R}^{1 \times p}$. Note that $z_f \sim \mathcal{N}(0, I_{c \times c})$, let $z_i$ be the $i$-th entry of $z$, we have

$$z_i = \sigma_f \sum_{k=1}^c \frac{\partial f_k}{\partial \theta_i} z_{f_k}. \quad (5)$$

Hence,

$$\mathbb{E}[z_i^2] = \sigma_f^2 \|\nabla_{\theta_i} f\|_2^2, \quad \mathbb{E}[z_i z_j] = \sigma_f^2 (\nabla_{\theta_i} f)^T \nabla_{\theta_j} f. \quad (6)$$

$\square$

**Proposition 2.** *For the cross-entropy loss, compared with Eq. (2), Eq. (3) induces noise $z \sim \mathcal{N}(0, \sigma_y^2 M)$ on $\nabla_f \ell(f, y)$, s.t., $M \in \mathbb{R}^{c \times c}$ $M_{i,i} = c(S_i - 1/c)^2 + (c-1)/c$, and $M_{i,j} = c \cdot S_i S_j - S_i - S_j$, if $i \neq j$. Note that the standard deviation of noise on the $i$-th entry is $\sigma_y \sqrt{c(S_i - 1/c)^2 + (c-1)/c}$.*

*Proof.* Dimension: $y \in \mathbb{R}^c, z_y \in \mathbb{R}^c, S \in \mathbb{R}^c, \nabla_f \ell \in \mathbb{R}^{1 \times c}, \nabla_S \ell \in \mathbb{R}^{1 \times c}, \nabla_f S \in \mathbb{R}^{c \times c}$.

Firstly, for the softmax function $S = S(f(x))$, we have the derivative matrix

$$\nabla_f S = \Lambda(S) - S \cdot S^T, \quad (7)$$

where $\Lambda(S)$ is a diagonal matrix with $S_i$ on its $i$-th diagonal element and $0$ otherwise.

For the cross-entropy loss, $\ell(f, y) = -\sum_{k=1}^c y_k \log S_k$. Let $\dot{\overline{S}}$ denote element-wise division, we have

$$
\begin{aligned}
\nabla_f \ell(f, y + \sigma_y z_y) &= \nabla_S \ell(f, y + \sigma_y z_y) \cdot \nabla_f S = -\left(\frac{y + \sigma_y z_y}{S}\right)^T \cdot \nabla_f S \\
&= -\left(\frac{y}{S}\right)^T \cdot \nabla_f S - \left(\frac{\sigma_y z_y}{S}\right)^T \cdot \nabla_f S \\
&= \nabla_f \ell(f, y) - \left(\frac{\sigma_y z_y}{S}\right)^T \cdot (\Lambda(S) - S \cdot S^T) \\
&= \nabla_f \ell(f, y) - \sigma_y \left(z_y - \sum_{k=1}^c z_{y_k} \cdot S\right)^T
\end{aligned}
\quad (8)
$$

Then it is equivalent to induce noise $z = -\sigma_y(z_y - \sum_{k=1}^c z_{y_k} \cdot S)$ on $\nabla_f \ell(f, y)$, whose $i$-th entry is $z_i = -\sigma_y(z_{y_i} - \sum_{k=1}^c z_{y_k} \cdot S_i)$. Note that $z_y \sim \mathcal{N}(0, I_{c \times c})$, then we can derive for $i = j$,

$$\mathbb{E}[z_i^2] = \sigma_y^2(1 - 2S_i + c \cdot S_i^2) = \sigma_y^2(c(S_i - 1/c)^2 + (c-1)/c), \quad (9)$$

and for $i \neq j$,

$$\mathbb{E}[z_i z_j] = \sigma_y^2(c \cdot S_i S_j - S_i - S_j). \quad (10)$$

$\square$

**Proposition 3.** *For the cross-entropy loss, compared with Eq. (1), Eq. (3) induces noise $z \sim \mathcal{N}(0, \sigma_y^2 M)$ on $\nabla_\theta \ell(f, y)$, s.t., $M \in \mathbb{R}^{p \times p}$ and $M_{i,j} = (\frac{\nabla_{\theta_i} S}{S})^T \frac{\nabla_{\theta_j} S}{S}, \forall i, j \in \{1, \cdots, p\}$, where $\dot{\overline{S}}$ denotes the element-wise division. Note that the standard deviation of noise on the $i$-th parameter $\theta_i$ is $\sigma_y \left\|\frac{\nabla_{\theta_i} S}{S}\right\|_2$.*

*Proof.* Dimension: $z_y \in \mathbb{R}^c$, $\nabla_\theta \ell \in \mathbb{R}^{1 \times p}$, $\nabla_S \ell \in \mathbb{R}^{1 \times c}$, $\nabla_\theta S \in \mathbb{R}^{c \times p}$, $\nabla_{\theta_i} S \in \mathbb{R}^c$.

For Eq. (3), the noisy gradient is

$$
\tilde{\nabla}_\theta \ell(f, y) = \nabla_\theta \ell(f, y + \sigma_y z_y) = \nabla_S \ell(f, y + \sigma_y z_y) \cdot \nabla_\theta S = -\left(\frac{y + \sigma_y z_y}{S}\right)^T \cdot \nabla_\theta S
$$
$$
= -\left(\frac{y}{S}\right)^T \cdot \nabla_\theta S - \left(\frac{\sigma_y z_y}{S}\right)^T \cdot \nabla_\theta S = \nabla_\theta \ell(f, y) - \left(\frac{\sigma_y z_y}{S}\right)^T \cdot \nabla_\theta S. \tag{11}
$$

The noise on $\nabla_\theta \ell(f, y)$ is $z = -\left(\frac{\sigma_y z_y}{S}\right)^T \cdot \nabla_\theta S$. Note that $z_y \sim \mathcal{N}(0, I_{c \times c})$, let $z_i$ be the $i$-th entry of $z$, we have

$$
z_i = -\sigma_y \sum_{k=1}^c \frac{\partial S_k}{\partial \theta_i} \frac{z_{y_k}}{S_k} = -\sigma_y \sum_{k=1}^c \frac{\nabla_{\theta_i} S_k}{S_k} z_{y_k}. \tag{12}
$$

Hence,

$$
\mathbb{E}[z_i^2] = \sigma_y^2 \left\| \frac{\nabla_{\theta_i} S}{S} \right\|_2^2, \quad \mathbb{E}[z_i z_j] = \sigma_y^2 \left( \frac{\nabla_{\theta_i} S}{S} \right)^T \frac{\nabla_{\theta_j} S}{S}. \tag{13}
$$

$\square$

# B MORE DETAILS ON EXPERIMENT SETUP

## B.1 CIFAR-10 AND CIFAR-100

**The backbone and general training hyperparameters.** In all experiments on CIFAR-10 and CIFAR-100, we train wide ResNet-28-2 (Zagoruyko & Komodakis, 2016) for 300 epochs using the SGD optimizer with learning rate 0.001, momentum 0.9, weight decay $5 \times 10^{-4}$, and a batch-size of 128. Standard data augmentation is applied, including per-pixel normalization, horizontal random flip and $32 \times 32$ random crop after padding with 4 pixels on each side. The criterion for setting the training hyperparameters includes 1) all methods should converge (the training accuracy converges), 2) all methods share the same general training hyperparameters for a fair comparison.

**Method-specific hyperparameters.** The backbone is not unified in previous papers and we reimplement all methods in the same backbone for a fair comparison. Regarding this, we may not directly follow the default hyperparameters. Following Zhang & Sabuncu (2018), we use $5k$ noisy samples (10% of the training data) as the validation set to tune method-specific hyperparameters. We then train the model on the full training set and report the test accuracy at the last epoch.

- **SLN/SLN-MO/SLN-MO-LC (ours).** We tune $\sigma \in \{0.1, 0.2, 0.5, 1\}$. On CIFAR-10, we use $\sigma = 1$ for symmetric noise and $\sigma = 0.5$ otherwise; On CIFAR-100, we use $\sigma = 0.1$ for instance-dependent noise and $\sigma = 0.2$ otherwise. The momentum model is introduced with hyperparemeter 0.999 without tuning. The label correction (LC) is applied after convergence of training with SLN. All models are trained for 300 epochs and we introduce LC at the 250th epoch without tuning beacuse the training accuracy does not increase much after the 250th epoch. In this way, we do not increase the computation cost.

- **GCE** (Zhang & Sabuncu, 2018). GCE loss is applied as the training starts and there is a warm epoch after which truncated GCE loss is applied every 10 epochs. We tune the warm epoch in $\{0, 50, 100, 150, 200\}$ and use 50 for CIFAR-10, 150 for CIFAR-100. There is a hyperparameter $q$ for the GCE loss. We set $q = 0.7$ since it is used in all experiments on CIFAR-10 and CIFAR-100 in its original paper (Zhang & Sabuncu, 2018).

- **Co-Teaching** (Han et al., 2018b). The rate of selecting small-loss samples is linearly decreased from 1 to $1 - \varepsilon$ at the first 10 epochs, where $\varepsilon$ is the noise rate. This setting is used in all experiments in the original paper (Han et al., 2018b) and it works well in our setting.

- **PHuber-CE** (Menon et al., 2020). There is a hyperparameter $\tau$ that controls the gradient clipping. The original paper (Menon et al., 2020) uses $\tau = 2$ on CIFAR-10 and $\tau = 10$ on CIFAR-100, but the default setting does not work well in our experiments. Hence, we tune $\tau \in \{2, 5, 10, 30, 50\}$ and finally, on CIFAR-10, we use $\tau = 10$ for asymmetric noise and $\tau = 2$ otherwise; on CIFAR-100, we use $\tau = 30$.

- **LS** (Lukasik et al., 2020). There is a hyperparameter $\alpha$ that controls how much the label is smoothed. We tune $\alpha \in \{0.2, 0.5, 0.8\}$ and finally, on CIFAR-10, we use $\alpha = 0.5$ for asymmetric noise and $\alpha = 0.8$ otherwise; on CIFAR-100, we use $\alpha = 0.8$.

- **SIGUA** (Han et al., 2020). The hyperpramemeter $\gamma$ is a factor that is multiplied on the loss on 'bad' samples. We tune it in $\{0.01, 0.001, 0.0001\}$ and finally use $\gamma = 0.001$ in all experiments.

- **DivideMix** (Li et al., 2020). We tune the warm-up epoch and $\lambda_u$ - the weight for unsupervised loss. In the official implementation, the warm-up epoch is 10 on CIFAR-10 and 30 on CIFAR-100. The default hyperparameters do not work well in our experiments (we observe a decrease of test accuracy after warm-up). Hence we tune the warm-up epoch in $\{10, 30, 50, 100\}$ but the performance does not improve compared with the default settings. Hence we use the default warm-up and tune $\lambda_u \in \{0, 1, 5, 10, 25\}$. Finally, we obtain impressive results with $\lambda_u = 0$ in all experiments.

## B.2 Clothing1M

**The backbone and general training hyperparameters.** On Clothing1M, following the common setting (Patrini et al., 2017), we train an Imagenet-pretrained ResNet-50 using the SGD optimizer with momentum 0.9, weight decay $10^{-3}$ and batchsize 32. The initial learning rate is $10^{-3}$ and decreased to $10^{-4}$ after 5 epochs. We use standard data augmentation with per-pixel normalization, horizontal random flip and $224 \times 224$ random crop. Note that DivideMix uses the same backbone ResNet-50 but a different training schedule, and we follow its official implementation released on the GitHub.

**Method-specific hyperparameters.** Since the backbone ResNet-50 is used by default for most published results (Patrini et al., 2017; Li et al., 2020), we can easily follow the default hyperparameters.

- **SLN/SLN-MO/SLN-MO-LC** (ours). Following previous methods (Patrini et al., 2017; Li et al., 2020), the validation set cobtaining $14k$ clean samples is adopted to tune our hyperparameters. We tune $\sigma \in \{0.1, 0.2, 0.5\}$ and choose $\sigma = 0.2$. The momentum model is implemented with hyperparemeter 0.999 without tuning. We fix the overall training epoch as 10 and tune the epoch for applying label correction in $\{1, 5, 9\}$. Finally, we apply label correction after the first epoch in all experiments.

- **Forward/Backword** (Patrini et al., 2017). The results is reproduced by reimplementing the method exactly following hyperparameters in Patrini et al. (2017).

- **Co-Teaching** (Han et al., 2018b). On Clothing1M, the estimated noise rate is around 0.4 (Xiao et al., 2015). Hence, we linearly reduce the rate of selecting small-loss samples from 1 to 0.6 in 10 epochs.

- **DivideMix** (Li et al., 2020). The result is reproduced from its official implementation.

## C More empirical results and discussions

### C.1 SLN enhances existing methods

With a detailed dissection on predictions of DNNs trained with SLN (Section 3.3), we have shown that SLN can boost popular robust training methods such as label correction and sample selection. In this section, we verify this by integrating SLN with the following methods.

- Co-teaching (Han et al., 2018b). It uses co-training and sample selection. Two modes select small-loss samples to train each other.

- Stochastic integrated gradient underweighted ascent (SIGUA) (Han et al., 2020). It adopts gradient descent on good data as usual, and learning-rate-reduced gradient ascent on bad data.

- DivideMix (Li et al., 2020). It combines co-training of two models, sample selection based on the loss, label correction/guessing based on semi-supervised learning MixMatch (Berthelot et al., 2019), and other techniques including regularization, augmenting each image twice.

Table 4: Testing accuracy (mean±std in 5 runs) on CIFAR-10. SLN consistently improves many robust learning methods. All methods are fairly compared using the same backbone wide ResNet-28-2 and training hyperparameters (Appendix B).

| Method | Symmetric | Asymmetric | Dependent | Open-Set |
|---|---|---|---|---|
| Co-Teaching | 82.37±0.32 | 79.61±1.06 | 76.69±0.66 | 85.21±0.35 |
| SLN-Co-Teaching | 84.22±0.43 | 87.79±0.17 | 80.37±0.22 | 90.37±0.32 |
| Improvement | **+1.85** | **+8.18** | **+3.68** | **+5.16** |
| SIGUA | 83.76±0.67 | 78.24±1.41 | 76.67±0.97 | 86.70±0.62 |
| SLN-SIGUA | 84.27±0.41 | 87.65±0.94 | 80.09±0.68 | 90.38±0.24 |
| Improvement | **+0.51** | **+9.41** | **+3.42** | **+3.68** |
| DivideMix | 90.38±0.34 | 87.88±0.45 | 82.21±0.37 | 90.49±0.62 |
| SLN-DivideMix | 90.87±0.28 | 89.31±0.39 | 82.86±0.41 | 91.65±0.59 |
| Improvement | **+0.49** | **+1.43** | **+0.65** | **+1.16** |

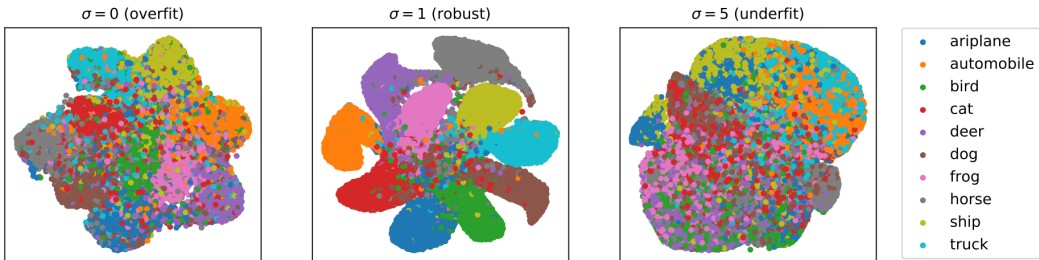

Figure 7: The t-SNE visualization of features on the model's penultimate layer.

Specifically, we use models trained with SLN as the initialization of these methods. As shown in Table 4, all methods obtain consistent improvement when integrated with SLN. Moreover, with SLN as initialization, we do not need to tune the warm-up phase for methods like DivideMix, because we can train with SLN until convergence. In contrast, without SLN, we need to carefully warm up the model so that it learns enough correct patterns and does not memorize too much noise. Note that better results for DivideMix are reported in the original paper with a different backbone and carefully scheduled learning rate. We focus on fairly comparing the robustness of all methods: in all experiments, we train the same backbone wide ResNet-28-2 for 300 epochs without learning rate change. Detailed training settings are presented in Appendix B.

## C.2    T-SNE VISUALIZATION OF FEATURES ON THE MODEL'S PENULTIMATE LAYER

In Fig. 7, we show the t-SNE visualization of features on the model's penultimate layer, taking all training samples as input. We visualize the embedding on CIFAR-10 with symmetric noise since the noise yields the most severe damage to the generalization and SLN provides a significant improvement. Fig. 7 shows that the model trained with SLN can yield a better embedding. It also demonstrates overfitting and underfitting when $\sigma$ is too small or too large.

## C.3    TEST ACCURACY W.R.T. TRAINING EFFICIENCY

In Fig. 8, we visualize the training efficiency and test accuracy. The figure clearly illustrates the superior generalization performance and high efficiency of our methods.

## D    LABEL CORRECTION

**The convergence issue in label correction.**    When using the model's prediction to correct noisy labels, we find a convergence issue such that the test accuracy decreases, as shown in Fig. 10. The convergence issue is also reported in Arazo et al. (2019), but it has not been widely discussed. In

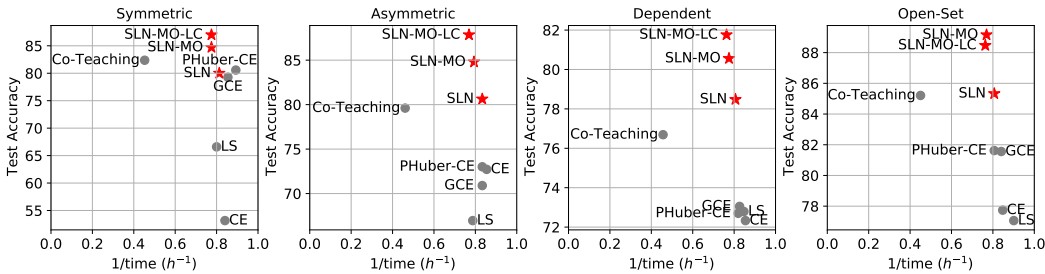

Figure 8: Test accuracy w.r.t. training efficiency (1/time) on CIFAR-10.

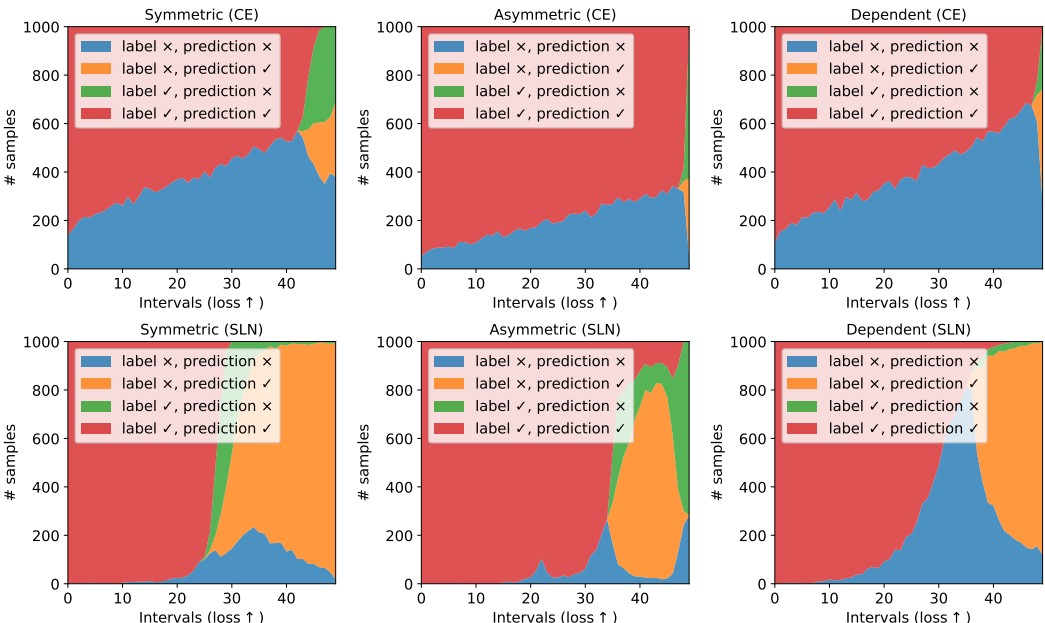

Figure 9: Training samples are sorted in ascending order of loss, uniformly divided into 1000 samples per interval, and dissected according to the correctness of the given label and the prediction. This figure is *exactly the same* as Fig. 4 in the main paper. We present it here for your convenience since we refer to the figure in Section D.

this section, we provide an intuitive hypothesis. We can analyze the effect of label correction on the four types of samples, as shown in Fig. 9.

- 1) label $\times$, prediction $\times$. Label correction can not correct these samples, yet may even make the case worse because the prediction error should be easier for the model to overfit compared to given noisy labels.

- 2) label $\times$, prediction $\checkmark$. These samples benefit from label correction.

- 3) label $\checkmark$, prediction $\times$. Label correction is harmful for these samples.

- 4) label $\checkmark$, prediction $\checkmark$. Label correction does not significantly impact these samples because both the prediction and the label are correct.

In a word, there exist samples of case 1) and case 3) that are not desired in label correction, but they are unavoidable since we do not expect a model with $100\%$ prediction accuracy. In label correction, when trained with modified labels obtained from wrong predictions, the model may accumulate its own error due to a positive feedback: yielding worse predictions after training on prediction errors. Therefore, we hypothesize that samples of case 1) and case 3) result in the convergence issue.

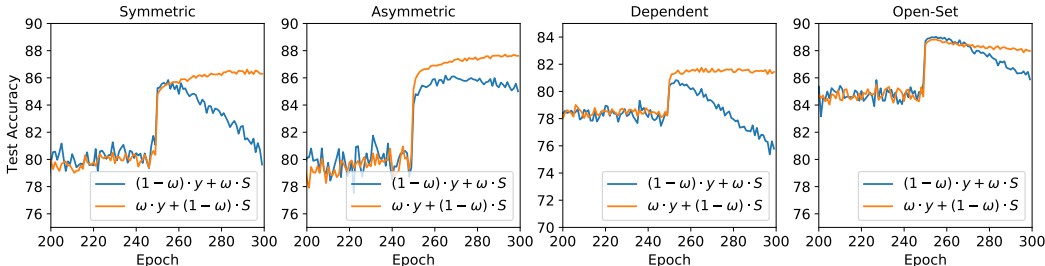

Figure 10: The test accuracy on CIFAR-10, averaged in 5 runs. We zoom in to focus on the effects of label correction applied in the last 50 epochs. The open-set noise involves samples that do not belong to any class considered in the task, for which label correction can not address.

**How do we assign weights in label correction?** We use $y_{correction} = \omega \cdot y + (1 - \omega) \cdot S$, as opposed to $y_{correction} = (1 - \omega) \cdot y + \omega \cdot S$, where $\omega$ is an instance-dependent weight positive correlated with the loss. We observe that our scheme mitigates the convergence issue, as shown in Fig. 10. Our intuition that - samples that need correction have large losses - is consistent with the method of Arazo et al. (2019), but this does not mean that the weight on $S$ should be positive correlated with the loss. We can consider the following cases.

- For small-loss samples, we have $S \approx y$, as illustrated by the red region in Fig. 9. *Label correction does not affect these samples much regardless of the weight*. Hence, we simply need to consider the effects of label correction on large-loss samples.

- Large-loss samples of case 2) can benefit from label correction, as illustrated by the orange region in Fig. 9. However, in this case, *a higher loss does not mean that it requires a higher weight on $S$ for label correction* [4].

- *There exist large-loss samples of case 1) and case 3) for which label correction can be harmful*, as has been discussed in the above paragraph and illustrated by the blue and green regions in Fig. 9.

Therefore, we assign a small weight of $1 - \omega$ on $S$ for large-loss samples. In this way, samples of case 2) still benefit from label correction, while we mitigate the undesired effects on other large-loss samples for which label correction can be harmful. The effectiveness of our scheme is verified in Fig. 10.

## E  THE CONVERGENCE

In Fig. 2, the visualizations of loss landscapes show that the model trained with SLN converges to a solution that has small SGD noise. The center point on the visualized landscape (i.e., the loss of the given model) is a local minimum. From Fig. 2 (d), we observe that the minimum has the following properties.

- The gradient around the minimum is small since it is flat.
- The predictions do not approach one-hot labels because the loss at the local minimum is high. As shown in Fig. 3, the prediction probabilities are much lower than 1.

With the above two properties, Proposition 3 implies that around the flat minimum illustrated in Fig. 2 (d), the noise on gradients is small. Therefore, the model can converge in the local flat minimum.

---

[4]For example, considering two samples with the wrong label $y_1 = y_2 = [1, 0, 0]$ and the latent true label $[0, 1, 0]$. Imaging the predictions are $S_1 = [0.4, 0.6, 0]$, $S_2 = [0.3, 0.4, 0.3]$. Then the cross-entropy loss is $\ell(y_1, S_1) < \ell(y_2, S_2)$, while for the weight on $S$, we want $w_1 > w_2$ because $S_1$ is more correct compared with $S_2$ (the second entry $0.6 > 0.4$). This example implies that for samples that can benefit from label correction, a higher loss does not mean that it requires a higher weight on $S$ for label correction.

