# OpenReview forum: "Noise against noise: stochastic label noise helps combat inherent label noise"
_ICLR.cc/2021/Conference — ICLR 2021 Spotlight_

### Official Review · AnonReviewer3 · 2020-10-25
**Very interest idea, but the empirical studies are not sufficient**

**Rating:** 6
**Confidence:** 5

**Review:**

This paper studies the noisy stochastic gradient descent algorithm in noisy label learning. Concretely, the authors added Gaussian noise on the labels rather than the gradient itself. By comparing different noisy SGD algorithms, the authors demonstrate that the proposed SLN algorithm not only help the model to escape from sharp local minima, but also help it to be over confidential. Finally, the authors demonstrate that SLN outperforms some classical SGD methods.

Pros:
- The proposed idea is novel to me. Theoretically, the proposed SLN framework jointly enjoys the ability to escape from sharp saddle points and make the prediction smooth. The authors provide a new perspective to develop robust learning algorithms.
- It seems that SLN method can be integrated with many state-of-the-art noisy label learning models.
- The experimental results in Figure 3 is very promising. The small loss samples are generally clean, which may help improve the performance of many sample-selection based approaches.

Cons:
- I have one main concern. While Figure 3 shows very good results, I noticed that quantitative results are far away from state-of-the-art models. Compared to state-of-the-art models, such as DivideMix, SLN demonstrates far lower accuracy. For example, on CIFAR-10, Asymmetric noisy with 40% noise, the accuracy of DivideMix is 92~93.4% and SLN-MO-LC is 87.85%. Although state-of-the-art performance is not the most essential for me, I think the authors require more experimental exploration. Since SLN is a rather flexible method, it can be integrated with many state-of-the-art models and I believe the performance would be competitive or at least at the same level as the SOTA models.

Minor comments:
- While Figure 3 shows the clean sample ratio of converged models w.r.t different loss intervals, what would it be after the first few epochs?
- From my point of view, SLN is actually a new strategy of label smoothing (or not?). May the authors explain the superiority of SLN to conventional label smoothing methods?

Overall, I think this paper brings an interesting idea to the community, but the experiments are not enough for me.

---

> ### Author Response · Authors · 2020-11-16
> **To Reviewer 3. We present more results and discussions according to the comments.**
>
> Thanks for the comments.
>
> 1. **More experimental explorations on the effectiveness of SLN.**
> Yes, SLN is a rather flexible method that is promising to improve existing methods. In the updated paper, we add Table 4 in Appendix C. We integrate SLN with three methods, including Co-Teaching, SIGUA, DivideMix, showing that SLN consistently improves these methods.
> We compare all methods in a fair setting: training the same backbone wide ResNet-28-2 for 300 epochs without learning rate change. Though a more powerful backbone and subtle learning rate schedule may yield better results, the additional results in Table 4 should be sufficient to verify the effectiveness of SLN: the methods obtain consistent improvement when integrated with SLN.
>
> 2. **The clean sample ratio w.r.t different loss intervals after the first few epochs.**
> According to the memorization effect (Arpit et al., 2017), DNNs tend to learn simple and correct patterns first before memorizing noise. Therefore, after the first few epochs, we still expect a higher clean sample ratio for the small-loss part. However, since the prediction accuracy is low after the first few epochs, there shall be many correctly or wrongly labeled samples with wrong predictions, i.e., the overall blue and green regions should be larger compared with the converged model.
> [Arpit et al., 2017] A closer look at memorization in deep networks. ICML 2017.
>
> 3. **The superiority of SLN to conventional label smoothing (LS).**
> SLN is not simply a new strategy of label smoothing. Please refer to Section 3.2 and Figure 3 in the updated paper, where we present a direct comparison between SLN and LS. The effects of SLN are very different from LS.
> $\bullet$ LS introduces a fixed and biased modification on labels to reduce overconfidence. Throughout the training, the labels are fixed.
> $\bullet$ SLN induces SGD noise that helps 1) escaping from sharp minima and 2) preventing overconfidence. The effects rely on random perturbations induced in each training step, as justified by Propositions 1-3 and Claims 1-2.
> For example, around a sharp minimum, SLN yields SGD noise with high variance and ‘random directions’ so that the noise helps escape. While for LS, the fixed labels do not induce dynamic perturbations so that the training always follows the direction of gradient descent.

---

### Official Review · AnonReviewer2 · 2020-10-25
**Interesting work on SGD noise and label noise, but many unclear parts need to be clarified**

**Rating:** 5
**Confidence:** 4

**Review:**

This paper studies learning robust models with noisy labels. The authors argue that a specific SGD noise induced by stochastic label noise (SLN) can mitigate the effect of label noise. But the common SGD noise cannot achieve this. Then they apply the proposed SLN induced SGD noise to the existing label-correction methods for noisy-label learning and provide some experimental results.

Pros:

-The paper provides an interesting view of SGD noise in the lens of noisy labels. They claim that common SGD noise does not endow much robustness against label noise, but using a variant SGD noise by label perturbations can improve the generalization and boost existing robust training methods.

Cons:

-Learning with noisy labels is a hot research area as reviewed in the related work section. It seems that the selected baselines in the experiments are not representative and state-of-the-art methods. For example, Yu et al. (2019) improves co-teaching Han et al. (2018) and should be compared instead.

And some representative regularization based methods should also be compared:

-Mixup: Beyond Empirical Risk Minimization, Zhang et at., ICLR 2018

-Virtual adversarial training: a regularization method for supervised and semi-supervised learning, Miyato et al., TPAMI 2019,

-SIGUA: Forgetting May Make Learning with Noisy Labels More Robust, Han et al., ICML 2020

since they are more related to the essence of training networks as claimed in the paper.

-In Claim 1 and 2, it is said that with SGD noise induced by SLN training is difficult to converge in some cases. How to guarantee convergence of the proposed algorithm? Some convergence analysis under reasonable assumptions may be helpful.

-It is claimed in the paper that training without SGD noise under label noise can converge to sharp minima, and SLN helps escape from the sharp minima. It is not very intuitive. Could the authors explain more on it, maybe adding some citations or experimental results could be helpful.

-It is still unclear to me how to tune the standard deviation sigma in practice, which should be an important factor that affects the performance.

-The clarity of the paper could be improved, for example, adding brief proof sketches to the theorems may help for better understanding.

Overall, the paper provides some interesting analysis of SGD noise and label noise, but many unclear parts need to be clarified.

---

> ### Author Response · Authors · 2020-11-16
> **To Reviewer 2. We present additional empirical results accordingly and further clarify our analysis and claims.**
>
> Thanks for the comments.
>
> 1. **More results to verify the effectiveness of SLN.**
> In the updated paper, we add Table 4 in Appendix C. We integrate SLN with three methods, including Co-Teaching, SIGUA, DivideMix, showing that SLN consistently improves these methods. We do not expect vanilla SLN to achieve state-of-the-art results compared with many integrated methods. Still, SLN can be a promising option in the family of robust learning methods. It can improve existing methods, as appreciated by Reviewer 3. Table 4 verifies this point. Please refer to the common response for a summary of the results.
>
> 2. **Illustrations of sharp/flat minima.**
> We add Figure 2 with visualizations of sharp/flat minima. The figure supports our claim very well. The model trained with SLN converges to a flat minimum, while the model trained with CE converges to a sharp minimum. Please refer to the common response and Figure 2 in the updated paper for more details.
>
> 3. **The convergence.**
> Please refer to the common response.
>
> 4. **How to tune the standard deviation $\sigma$.**
> We briefly discuss the hyperparameter optimization in Section 4 and provide a detailed discussion in Appendix B. $\sigma$ is tuned in {0.1, 0.2, 0.5, 1}. On CIFAR10 and CIFAR-100, following Zhang & Sabuncu (2018), we use 5k noisy samples (10% of the training data) to tune $\sigma$. On Clothing1M, following the standard setting (Patrini et al., 2017), we use the clean validation set to tune $\sigma$.
> Moreover, we add Figure 6 in Section 4 to show an ablation study on $\sigma$ on CIFAR-10.

---

> > ### Comment · AnonReviewer2 · 2020-11-16
> > **Thanks for adding more experimental results and updating the paper!**
> >
> > Thanks for adding the experimental results of integrating SLN with several sota label-noise learning methods,
> > and the illustrations of sharp/flat minima.
> > The results support the claim in the paper well.
> >
> > This reminds me of some interesting explanations of overfitting from the information-theoretic view.
> > It is known that the CE loss can be decomposed into several terms
> > (see this paper Sec. 4 Eq. 2: Emergence of Invariance and Disentanglement in Deep Representations,  JMLR 2018)
> > and the only negative term which relates to how much information about the labels is memorized in the weights may explain the poor generalization.
> > Several works follow this and propose methods to limit the label information to improve generalization in label-noise learning,
> > see
> > Improving Generalization by Controlling Label-Noise Information in Neural Network Weights, arXiv 2020
> > Artificial Neural Variability for Deep Learning: On Overfitting, Noise Memorization, and Catastrophic Forgetting, arXiv 2020
> >
> > It would be interesting to see if the proposed method can also be supported by this.

---

> > > ### Author Response · Authors · 2020-11-16
> > > **Thanks for acknowledging that our additional results support the claim well. Thanks for pointing out the reference, which further supports our method.**
> > >
> > > Yes, the work (Achille & Soatto, 2018) supports our method. We update the paper and add a discussion in the first paragraph of Section 3.2.
> > >
> > > For our SLN, we analyze and illustrate the effect of leading to flat minima and show better generalization performance. The work (Achille & Soatto, 2018) supports our method because it theoretically bridges the gap between flat minima and better generalization performance: 1) Flat minima have low information between the weights and the data (Section 4.3 and Proposition 4.3 in their paper); 2) low information reduces overfitting (Section 4 Eq. (2) and the discussions in their paper). Thanks for pointing out the methods (Harutyunyan et al., 2020;
> > > Xie et al., 2020) that follow the finding in Achille & Soatto (2018). We cite them in the updated paper.
> > >
> > > [Achille & Soatto, 2018] Emergence of Invariance and Disentanglement in Deep Representations, JMLR, 2018.
> > > [Harutyunyan et al., 2020] Improving Generalization by Controlling Label-Noise Information in Neural Network Weights. ICML, 2020.
> > > [Xie et al., 2020] Artificial Neural Variability for Deep Learning: On Overfitting, Noise Memorization, and Catastrophic Forgetting. arXiv 2020.

---

> > > ### Author Response · Authors · 2020-11-19
> > > **To Reviewer 2. Thanks for your positive feedback on the updated paper. Please feel free to discuss if there is anything else to clarify.**
> > >
> > > Looking forward to your updates.
> > >
> > > Sincerely,
> > > Paper568 Authors.

---

### Official Review · AnonReviewer1 · 2020-11-02
**The paper is original and brings consistent improvements. However, the connection of the claims to the performance improvement remain empirically unsubstantiated. Furthermore, the paper can improve its clarity.**

**Rating:** 7
**Confidence:** 4

**Review:**

**Summary** The paper tackles the problem of training under noisy labels. It proposes adding random  zero-mean Gaussian noise to the labels during training. It is shown that such a noise induces a variable gradient noise which adaptively increases 1) when the learnt network function has higher curvature around training points and 2) when the output prediction has lower entropy (higher confidence) for the training points. It is claimed that the former property helps avoiding sharper minima which generally improves generalization. The second property avoids overfitting to noise in similar fashion to label smoothing.

**Quality** The paper is well written. The “related works” section is quite thorough but concise when covering the field of learning under label noise. Certain parts such as the tSNE plot (Figure 5) and the time-vs-accuracy plot (Figure 6) do not seem to be central to the paper and are not informative. Instead, certain parts could have been discussed more thoroughly (see the detailed technical comments below).

**Clarity** The claims and contributions are generally clear from the paper. The reasoning behind certain claims could be better clarified. Also, details are missing on the hyperparameter optimization of the baselines. Details come below.

**Originality** The method is close to other works in the analysis of noisy gradients for better generalization as well as the usage of label smoothing and random label perturbation [a] for generalization and learning under label noise. However, to the knowledge of the reviewer, this work combines the ideas of the two directions in a coherent and original way.

**Significance** The experiments are done on several setups and using 5 different independent runs for the baselines and three variants of the proposed method. The results show significant and consistent improvements. However, certain experiments could be added to better support the detailed claims as opposed to merely reporting best final numbers.

**Major technical comments**

*Experiments*
1. Interesting and informative side experiments including 1) the separation of noisy and correctly labeled data when using SLN compared to label smoothing, and 2) the strength of SLN compared to unnoisy cross entropy loss indicating the suitability of SLN in identifying correctly labelled samples (low-loss regime) and correcting noisy labels (high-loss regime).
2. 5 different synthetic noise types are used for the experiments on CIFAR10 and CIFAR100.
3. the paper has two clear claims regarding the sharpness of the found local minimum and on overconfident predictions. While the experiments show clear improvements of the results across various settings and compared to different baselines, the connection of the improvements to the claims remain largely unsubstantiated. As such, the paper is missing direct experiments supporting the claims and/or shedding some light on them. Some suggestions are as follows:

     3.1. implement the noise as in proposition 3 i.e., directly applying the noise to the gradient. Then, one can modify the noise to decouple the two components and demonstrate individual contributions.

     3.2. quantitatively analyze the jacobian of the learnt network function and/or the sharpness of the local minimum with and without the added noise and for the different kinds of noise to directly investigate the first claim.
4. the performance of some of the baselines are low. Are the baselines reimplemented? How are the general training hyperparameters (learning rate, weight decay, batchsize, etc) and method-specific hyperparameters optimized? Is there a different set of optimized hyperparameters per baseline? What are them?
5. reading the end part of section 3.3 it seems that in SLN+MO+LC the label correction starts only after the full convergence of a SLN-only training. In light of this, how is that the increase in time complexity of SLN+MO+LC is negligible in Table 1? What is the stopping criteria for the initial training and then retraining with LC?

*Theory*
1. The paper misses to cite a relevant paper [a] that also randomizes the labels for avoiding overfitting and demonstrates better generalization. Regarding this, the paper should clearly acknowledge [a] when it comes to claiming the novelty of the noisy-label approach and also when it discusses the advantages of perturbing labels -- [a] discusses similarities to ensemble approaches. That being said, I believe the paper has enough originality on top of [a]: for instance [a] replaces provided labels with independent noise, does not experiment on learning under noisy labels and, the gradient noise analysis of the paper is complementary to [a].
2. When correcting the labels using SLN, why is the weight of the given label increases as the sample loss increases? Shouldn’t it be the opposite based on figure 3? I found that there are some discussions provided in appendix C. However, as this goes against the previously published work it deserves more formal discussion and corresponding experiments in the main paper.
3. From what I understand proposition 3 shows that functions that are smoother at training points will receive lower variance. If so, formal discussions are missing to connect smooth functions and flat minima.
4. As the training continues the loss tends to get smaller by getting the function closer to the given one-hot labels at the training data points. From proposition 3 it is argued that as this happens the noise in the gradient increases. This raises a caveat regarding convergence. A theoretical discussion and/or empirical observation are needed to study the convergence. For instance, does the variance of the model increase towards the end of the training or does it actually converge to a solution that is robust to the gradient noise (remains approximately unchanged in the functional space)?

**Overall** In the reviewer’s view, the paper has clear merits in bridging between the theory of gradient noise and label smoothing for learning under label noise, both empirically and theoretically. However, it can benefit from more clarity and additional informative experiments to better understand the effect of the proposed noise.

[a] "DisturbLabel: Regularizing CNN on the Loss Layer", CVPR 2016

**Post Rebuttal Update**
The authors address many of the concerns, 1) [a] is properly acknowledged in the revised version and novelty is not claimed on additional label noise in the text, 2) while quantitative studies are still absent for claims on sharp vs. flat minima, qualitative results are provided for convergence to "flatter" minima  3) connections between smooth functions and generalization is discussed 4) answers and updates regarding complexity and convergence are *somewhat* convincing. Thus, I am willing to increase the score from 6 to 7 and confidence from 3 to 4 as I believe the paper provides relevant and interesting theoretical arguments.

---

> ### Author Response · Authors · 2020-11-16
> **To Reviewer 1. According to the comments, we present more empirical results to verify the claims and polish the discussions to improve clarity.**
>
> Thanks for the comments. We move the t-SNE plot and the time-vs-accuracy plot to the appendix and discuss certain parts more thoroughly according to the comments (see the detailed response below).
>
> Experiments
>
> 1. No question.
>
> 2. No question.
>
> 3. **Direct experiments supporting the claims and/or shedding some light on them.**
> $\bullet$ For the sharp/flat minima, we add visualizations of landscapes in Figure 2. The figure supports our claim very well. More details can be found in the common response and the updated paper. The model trained with noise in Eq. (2) or Eq. (3) converges to flat minima.
> $\bullet$ For preventing overconfidence, histograms in Figure 3 (Figure 2 in the original submission) show that the prediction probabilities on both noisy and correctly labeled data are reduced.
>
> 4. **The hyperparameter optimization.**
> We update the paper and describe the hyperparameter optimization in detail in Appendix B. On CIFAR-10 and CIFAR100, all methods are reimplemented and fairly compared with 1) the same general training hyperparameters and 2) method-specific hyperparameters tuned. Following Zhang & Sabuncu (2018), we use 5k noisy samples (10% of the training data) to tune hyperparameters. On Clothing1M, published results share the same backbone ResNet-50. Hence, we easily implement the methods by following the suggested hyperparameters. The result of DivideMix is reproduced from its official implementation. More details can be found in Appendix B.
>
> 5. **The stopping criteria for the initial training and then retraining with label-correction (LC).**
> Yes, we should introduce LC after the convergence of a SLN-only training. The increase in time complexity of SLN+MO+LC is negligible because we apply LC at the 250th epoch without tuning and all models are still trained for 300 epochs in total. The reasons are: 1) we find that the training accuracy does not increase much in the last 50 epochs (indicating convergence); 2) we can avoid increasing the time complexity.
>
>
> Theory
>
> 1. **A relevant paper [a] "DisturbLabel: Regularizing CNN on the Loss Layer", CVPR 2016.**
> We update the paper to acknowledge [a] in the related works, the method section and when discussing the effects. Yes, our originality includes analysis of SGD noise variants and the effects, and experiments in learning with noisy labels.
>
> 2. **The weight in label-correction (LC).**
> We polish the discussion in Appendix D to make it clearer. Figure 10 provides empirical justifications. In summary, the reasons are as follows.
> $\bullet$ For small-loss samples, we have $S \approx y$. LC does not affect these samples much regardless of the weight.
> $\bullet$ Samples that can benefit from LC have large-loss, but for these samples, a higher loss does not mean that it requires a higher weight on the prediction S (an example is provided in the updated paper).
> $\bullet$ There exist large-loss samples for which label-correction can be harmful because the prediction accuracy is not 100%.
> Therefore, we add small weights on the prediction S for large-loss samples to correct the labels slightly. More detailed discussions can be found in Appendix D.
>
> 3. **Smooth functions and flat minima.**
> Yes, functions that are smoother at training points will receive a lower variance. Still, the smoothness is characterized w.r.t. change in model parameters $\theta$ rather than the input x, because the gradient is w.r.t. $\theta$ rather than x. Around a flat minimum, the gradient of loss w.r.t. $\theta$ is small. The loss is averaged on batches of samples in training and averaged on the whole training set when we visualize the loss landscapes in Figure 2.
>
> 4. **The convergence.**
> Please refer to the common response.

---

### Official Review · AnonReviewer5 · 2020-11-04
**Nice formalization of SGD noises variants with intuitive theoretical justification and comprehensive empirical results**

**Rating:** 7
**Confidence:** 3

**Review:**

Summary:
mitigate inherent label noise
This paper studies the effect of applying SGD noise on mitigating the inherent label bias which is common in real-world datasets. It introduces stochastic label noise (SLN), a variant of SGD noise induced by controllable label noise. It formalizes connections between SLN and two other existing SGD noise variants (Proposition 1-3). With such propositions, it shows that SLN can help the model to avoid sharp minima and prevent overconfidence (Claim 1-2). The experiments show that SLN helps improve generalization than baseline methods and can be further used to boost robust training methods on CIFAR10, CIFAR100, CLOTH-1M under five different types of noise settings. Apart from vanilla SLN, it further proposes momentum model (MO) and label correction (LC). Combining them together with SLN can further boost test accuracy for label-correction.


################################################

Reasons for score:
The paper is overall very well-written and gives theoretical insight on the connections of different SGD noise variants. It further provides comprehensive experiments both qualitatively and quantitatively to validate the effectiveness of its proposed SGD noise variant, SLN. The results show that SLN can simplify parameter tuning, producing superior results on label-correction without additional computational overheads.

################################################

Pros:

+very well-written and easy to follow in general

+great connection with as well as comprehensive discussions of related work

+comprehensive experiments along with good visualization

+the proposed method gives better performance without overhead and can be used to enhance existing methods

Cons:

-might be better to also give empirical evidence to support the claim of helping escaping sharp minimums. e.g. a visualization of gradient landscapes for CE and SLN, respectively.


################################################

Questions:

-As I mentioned in the cons, is there any empirical evidence of escaping sharp minima you observed to further support the theoretical finding?

-I see that in Fig 5, you give a qualitative visualization of using different sigma. Did you also do any quantitative ablation study on the hyperparameter sigma? How sensitive the results would be by choosing different sigma?

################################################

Post Rebuttal Update: the authors have well addressed my concerns, in particular (1) the additional visualization gives a good qualitative empirical evidence supporting the claim that SLN helps escaping sharp minima. (2) the search process for the hyperparameter $\sigma$ is very reasonable and makes the usage of SLN practical. I will keep my initial assessment and vote for accepting this paper.

---

> ### Author Response · Authors · 2020-11-16
> **To Reviewer 5. We present more empirical results and visualizations accordingly.**
>
> Thanks for the comments.
>
> 1. **Visualizations of landscapes.**
> We add Figure 2 with visualizations of sharp/flat minima. The figure supports our claim very well. The model trained with SLN converges to a flat minimum, while the model trained with CE converges to a sharp minimum. Please refer to the common response and Figure 2 in the updated paper for more details.
>
> 2. **Ablation study on the hyperparameter $\sigma$.**
> We add Figure 6 to show an ablation study on $\sigma$ on CIFAR-10 by changing the value of $\sigma$ around the best value found in our original experiments. Note that in the original submission, $\sigma$ is tuned in {0.1, 0.2, 0.5, 1}. On CIFAR10 and CIFAR-100, following Zhang & Sabuncu (2018), we use 5k noisy samples (10% of the training data) to tune $\sigma$. On Clothing1M, following the standard setting (Patrini et al., 2017), we use the clean validation set to tune $\sigma$.

---

> > ### Comment · AnonReviewer5 · 2020-11-19
> > **Thank you for the additional results! I have some concerns regarding the hyperparameter.**
> >
> > Thank you for providing detailed additional results! The visualization looks quite nice and it supports the claim nicely.
> >
> > I have some additional concerns regarding the hyperparameter based on the additional ablation study.
> > For the hyperparameter $\sigma$, it seems that SLN's performance is a bit sensitive to its value. Also, different $\sigma$ works best for different datasets (0.5, 1 for CIFAR10 and 0.1, 0.2 for CIFAR100). Is there an easy way to choose the best value other than trying many for each new dataset and type of noise? In particular, I notice that the baseline method z_f (on Figure 5), if finetuned properly for each type of noise, seems to achieve similar level of accuracy.

---

> > > ### Author Response · Authors · 2020-11-19
> > > **To Reviewer 5. Thanks for your feedback! Thanks for appreciating the additional results and visualizations!**
> > >
> > > Yes, we do have a guideline for tunning $\sigma$. Given a new dataset with unknown noise, we suggest quickly search the best $\sigma$ (denoted as $\sigma_{0}$) based on the binary search using the validation accuracy throughout training.  1) If we observe a decrease of validation accuracy at late stage of training, it implies overfitting and $\sigma<\sigma_{0}$. 2) Otherwise, we have $\sigma\geq\sigma_{0}$. Based on 1) and 2), we can conduct a binary search for $\sigma$, which can quickly find the best $\sigma$ especially if one would like to search $\sigma$ in a very detailed range. Finally, we would like to emphasize that tuning the hyperparameter for SLN is easy compared with many baselines since we simply need to tune $\sigma$. And in our paper, we achieve good results by simply tuning $\sigma$ in {0.1, 0.2, 0.5, 1}.
> > >
> > > Previously, both $z_f$ and SLN are not adopted in learning with noisy labels. Our originality includes adopting the techniques in combating noisy labels and notably, the analysis of the effects of SGD noise variants, which are supported by the empirical results and visualizations. Based on our analysis, $z_f$ also has the effects of escaping sharp minima; hence it endows robustness. SLN is better in terms of reducing overconfidence.

---

> > > > ### Comment · AnonReviewer5 · 2020-11-19
> > > > **Thank you for the additional details!**
> > > >
> > > > The additional details have addressed my concerns.

---

### Public Comment · ~Ehsan_Amid1 · 2020-11-10
**Please consider referencing/comparing to these more recent works**

I would like to point out that our work (Amid et al. 2019a) extends the Generalized CE loss (Zhang and Sabuncu 2018) by introducing two temperatures t1 and t2 which recovers GCE when t1 = q and t2 = 1. Our more recent work, called the bi-tempered loss (Amid et al. 2019b) extends these methods by introducing a proper (unbiased) generalization of the CE loss and is shown to be extremely effective in reducing the effect of noisy examples. Please consider referencing/comparing to these papers.

(Amid et al. 2019a) Amid et al. "Two-temperature logistic regression based on the Tsallis divergence." In The 22nd International Conference on Artificial Intelligence and Statistics (AISTATS), 2019.

(Amid et al. 2019b) Amid et al. "Robust bi-tempered logistic loss based on Bregman divergences." In Advances in Neural Information Processing Systems (NeurIPS), 2019.

---

### Author Response · Authors · 2020-11-16
**[Paper updated]. Common response to all the reviewers. Thanks for the very detailed constructive comments. We polish the paper with additional empirical results and discussions according to the comments.**

The analysis and empirical results presented in this paper should interest not only the subfield of learning with noisy labels but also a wide area of optimization/regularization. The additional visualizations of sharp/flat minima are very interesting and support our claims very well. The major updates in the paper are as follows.

1. **Figure 2 in Section 3.2: visualizations of sharp/flat minima.**
We visualize loss landscapes of converged models to verify our claim of escaping from sharp minima. The loss is averaged on all training samples. The landscapes are visualized using the technique in Li et al. (2018), which perturbs the model parameters in two directions on a normalized scale. Notably, the results exactly fit our analysis. (a): The model trained with CE converges to a sharp minimum. (b): Training with Eq. (1) yields a minimum with a higher loss, yet it is still sharp. (c)&(d): Consistent with our analysis, the model trained with Eq. (2) or Eq. (3) (SLN) converges to a flat minimum.
[Li et al., 2018] Visualizing the loss landscape of neural nets. NeurIPS 2018.

2. **Table 4 in Appendix C: more results showing that SLN improves existing robust learning methods.**
We integrate SLN with three methods, including Co-Teaching, SIGUA, DivideMix. We do not expect vanilla SLN to achieve state-of-the-art results compared with many integrated methods. Still, SLN can be a promising option in the family of robust learning methods. It can improve existing methods, as appreciated by Reviewer 3.
We compare all methods in a fair setting: training the same backbone wide ResNet-28-2 for 300 epochs without learning rate change. Though a more powerful backbone and subtle learning rate schedule may yield better results, the additional results in Table 4 should be sufficient to verify the effectiveness of SLN: the methods obtain consistent improvement when integrated with SLN.
We summarize the results here.

    |    |  Symmetric  |  Asymmetric  |  Dependent  |  Open-Set  |
    | :----: | :----: |  :----: |  :----: |  :----: |
    Co-Teaching | 82.37±0.32 | 79.61±1.06 | 76.69±0.66 | 85.21±0.35 |
    SLN-Co-Teaching | 84.22±0.43 | 87.79±0.17 | 80.37±0.22 | 90.37±0.32 |
    **Improvement** | **+1.85** | **+8.18** | **+3.68** | **+5.16** |
    SIGUA | 83.76±0.67 | 78.24±1.41 | 76.67±0.97 | 86.70±0.62 |
    SLN-SIGUA | 84.27±0.41 | 87.65±0.94 | 80.09±0.68 | 90.38±0.24 |
    **Improvement**  | **+0.51** | **+9.41** | **+3.42** | **+3.68** |
    DivideMix | 90.38±0.34 | 87.88±0.45 | 82.21±0.37 | 90.49±0.62 |
    SLN-DivideMix | 90.87±0.28 | 89.31±0.39 | 82.86±0.41 | 91.65±0.59 |
    **Improvement** | **+0.49** | **+1.43** | **+0.65** | **+1.16** |
3. **Figure 6 in Section 4: the ablation study on $\sigma$.**
SLN simply requires tuning $\sigma$, which is tuned in {0.1, 0.2, 0.5, 1}. On CIFAR10 and CIFAR-100, following Zhang & Sabuncu (2018), we use 5k noisy samples (10% of the training data) to tune $\sigma$. On Clothing1M, following the standard setting (Patrini et al., 2017), we use the clean validation set to tune $\sigma$.
We add Figure 6 to show an ablation study on $\sigma$ on CIFAR-10 by changing the value around the best one found in our original experiments. Notably, the figure implies that we can even achieve better results than reported in Table 1 with a more detailed hypermarameter search.

4. **The convergence.**
We add Appendix E to discuss the convergence. The visualizations of loss landscapes (Figure 2 in the updated paper) show that **the model trained with SLN converges to a solution that has small SGD noise**. Firstly, the center point on the visualized landscape (i.e., the loss of the given model) is a local minimum. From Figure 2 (d), we observe that the minimum has the following properties.
$\bullet$ The gradient around the minimum is small since it is flat.
$\bullet $ The predictions “do not approach one-hot labels” because the loss at the local minimum is high. As shown in Figure 3, the prediction probabilities are much lower than 1.
With the above two properties, Proposition 3 implies that around the flat minimum illustrated in Figure 2 (d), the noise on gradients is small. Therefore, the model converges in the local flat minimum.

5. **Please refer to the individual response for more detailed discussions.**

---

### Decision · Program_Chairs · 2021-01-07
**Final Decision**

**Decision:**

Accept (Spotlight)

**Comment:**

The paper studies the effect of explicitly introducing stochastic label noise into SGD updates, showing both theoretically and empirically that this can improve model performance on datasets with "inherent" label noise. The intuition is that this helps the model escape sharp local minima, where predictions may be overconfident.

Reviewers broadly found the work to be conceptually and theoretically interesting, and the empirical results are promising. The paper is thus well-posed to be of broad interest to the community.